# Cardiac fibroblasts regulate myocardium and coronary vasculature development in the murine heart via the collagen signaling pathway

Yiting Deng[1†], Yuanhang He[1,2†], Juan Xu[1], Haoting He[1], Manling Zhang[3], Guang Li[1]*

[1]Department of Cell Biology, University of Pittsburgh School of Medicine, Pittsburgh, United States; [2]Tsinghua University, Tsinghua medicine, School of Medicine, Beijing, China; [3]Vascular Medicine Institute, Department of Medicine, University of Pittsburgh, Pittsburgh, United States

## eLife Assessment

This study provides a comprehensive analysis of gene expression and bioinformatics data, offering **important** insights into the roles of fibroblasts in cardiac development. The large and well-analyzed single-cell RNA sequencing (scRNA-seq) dataset is **compelling** and a significant contribution to the field, and will be of broad interest to the scientific community.

**Abstract** The fibroblast (FB), cardiomyocyte (CM), and vascular endothelial cell (Vas_EC) are the three major cell types in the heart, yet their relationships during development are largely unexplored. To address this gap, we employed RNA staining of the FB marker gene *Col1a1* together with the CM marker gene *Actn2* and the Vas_EC marker gene *Cdh5* at various stages of mouse heart development. This approach enabled us to discern the anatomical pattern of cardiac FBs and identify approximately one EC and four CMs directly interacting with each FB. Molecularly, through the analysis of single-cell mRNA sequencing (scRNA-seq) data, we unveiled collagen as the top signaling molecule derived from FBs influencing CM and Vas_EC development. Subsequently, we used a *Pdgfra-CreER* controlled diphtheria toxin A (DTA) system to ablate the FBs at different stages. We found that the ablation of FBs disrupted myocardium and vasculature development and led to embryonic heart defects. Using scRNA-seq, we further profiled the ablated hearts and identified molecular defects in their ventricular CMs and Vas_ECs compared to control hearts. Moreover, we identified a reduction of collagen in the ablated hearts and predicted collagen as the major signaling pathway regulating the differentially expressed genes in the ablated ventricular CMs. Finally, we performed both short-term and long-term FB ablation at the neonatal stage. We found that short-term ablation caused a reduction in collagen and Vas_EC density, while long-term ablation may induce compensatory collagen expression without causing heart function reduction. In summary, our study has identified the function of FBs in regulating myocardium and vasculature development in the mouse heart and implicated an important role for the collagen pathway in this process.

## Introduction

Fibroblast (FB) is one of the most abundant cardiac cell types and plays important roles in normal heart function and pathological heart remodeling at the adult stage (*Hall et al., 2021*; *Tallquist and*

**\*For correspondence:**
guangli@pitt.edu

[†]These authors contributed equally to this work

**Competing interest:** The authors declare that no competing interests exist.

*Molkentin, 2017*). The FB composition in the heart has been analyzed with multiple approaches such as flow cytometry, with a focus on the left ventricle (*Zhou and Pu, 2016*; *Pinto et al., 2016*). However, the anatomical location of FBs in the heart along the developmental progression is still unclear. The heart in mice starts to develop into a four-chambered structure at E9.5. The atrial and ventricular chambers are connected by the atrioventricular canal (AVC), a transient structure that develops into the septum and valve cells (*de Vlaming et al., 2012*; *O'Donnell and Yutzey, 2020*). Although the main FB population does not develop until E13.5 (*Tallquist, 2020*), FB-like cells with the expression of *Col1a1* and other FB genes start to develop in the AVC at E9.5 (*O'Donnell and Yutzey, 2020*).

CM and Vas_EC are the other two major cell types in the heart. Vas_EC developed at about the same time as FB, while CM developed before the appearance of FB at the chambers. FB has been shown to couple with CM through gap junctions and promote their maturation in human three-dimensional micro-tissue consisting of FB, CM, and EC (*Giacomelli et al., 2020*). In mice, epicardium-derived cells, mostly FBs, have been reported to serve as guidepost cells by secreting the chemokine *Slit2*, guiding Vas_EC to the correct anatomical locations and promoting their maturation (*Quijada et al., 2021*). However, the lack of direct assessment of FB function in vivo prevents a comprehensive understanding of its role in regulating CM and Vas_EC development.

To study the function of a cell type rather than a gene, toxin genes such as NTR and DTA have been engineered into conditional ablation systems to eliminate specific groups of cells (*Ivanova et al., 2005*; *Curado et al., 2008*). The DTA system has been used in mice to ablate cardiac progenitor cells and cardiomyocytes at early developmental stages (*Sturzu et al., 2015*). Recently, a group of prolif-erating FBs with the expression of *Postn* was identified in neonatal hearts. Further elimination of this cell population using the DTA ablation system showed its importance in promoting cardiomyocyte maturation (*Hortells et al., 2020*). Additionally, the DTA system has been used to ablate FBs at the adult stage to assess their function in adverse cardiac remodeling (*Kuwabara et al., 2022a*; *Kaur et al., 2016*). However, the function of the primary FB population during embryonic and neonatal stages remains unclear.

One of the major functions of FB is to synthesize and secrete extracellular matrix (ECM) proteins, which influence the behavior of other cell lineages (*Tallquist and Molkentin, 2017*; *Ivey and Tallquist, 2016*). The ECM is heterogeneous and composed of various types, such as collagen, fibronectin, elastin, and laminin, each with different subtypes. For example, collagen has 28 genes in vertebrates (*Frantz et al., 2010*). The ECM plays important roles in heart development and regeneration. For instance, *hyaluronan and proteoglycan link protein 1 (Hapln1)* is critical in regulating myocardial compaction and heart regeneration in zebrafish (*Sun et al., 2022*). ECM has also been found to regu-late mouse CM proliferation through β1-integrin signaling in a cell co-culture assay (*Ieda et al., 2009*). Moreover, collagen has been shown to be essential for heart regeneration in adult zebrafish and mice (*Simões et al., 2020*). However, the heterogeneity and function of ECM genes in mouse heart devel-opment remain unclear.

In this study, we identified the anatomical patterns of cardiac FB, CM, and Vas_EC during embry-onic development. We further examined the ligand-receptor interactions among these cell types and identified collagen as the top signaling pathway. Additionally, we investigated FB function at different embryonic stages using a cell ablation system and analyzed the resulting molecular defects with scRNA-seq. Finally, we performed short-term and long-term FB ablation at the neonatal stage and found that FBs are dispensable for normal heart function development. In summary, this study provides a comprehensive understanding of the role of cardiac fibroblasts in myocardium and coro-nary vasculature development during embryonic and neonatal stages.

## Results

### Cellular analysis of cardiac FB anatomical location at different stages

To understand the spatial distribution of cardiac FBs along the developmental progression, we stained the FB marker gene *Col1a1* on heart sections at different stages from E11.5 to P3. We observed a strong expression of *Col1a1* in epicardial cells at all analyzed stages (*Figure 1A*). Inside the heart, we found *Col1a1*-positive cells enriched at the boundary of the atrial and ventricular regions in E11.5 and 12.5 hearts (*Figure 1A*, *Figure 1—figure supplement 1*), indicating their presence in the atrio-ventricular canal (AVC). By E13.5 and 14.5, the AVC cells had developed into valve structures, and

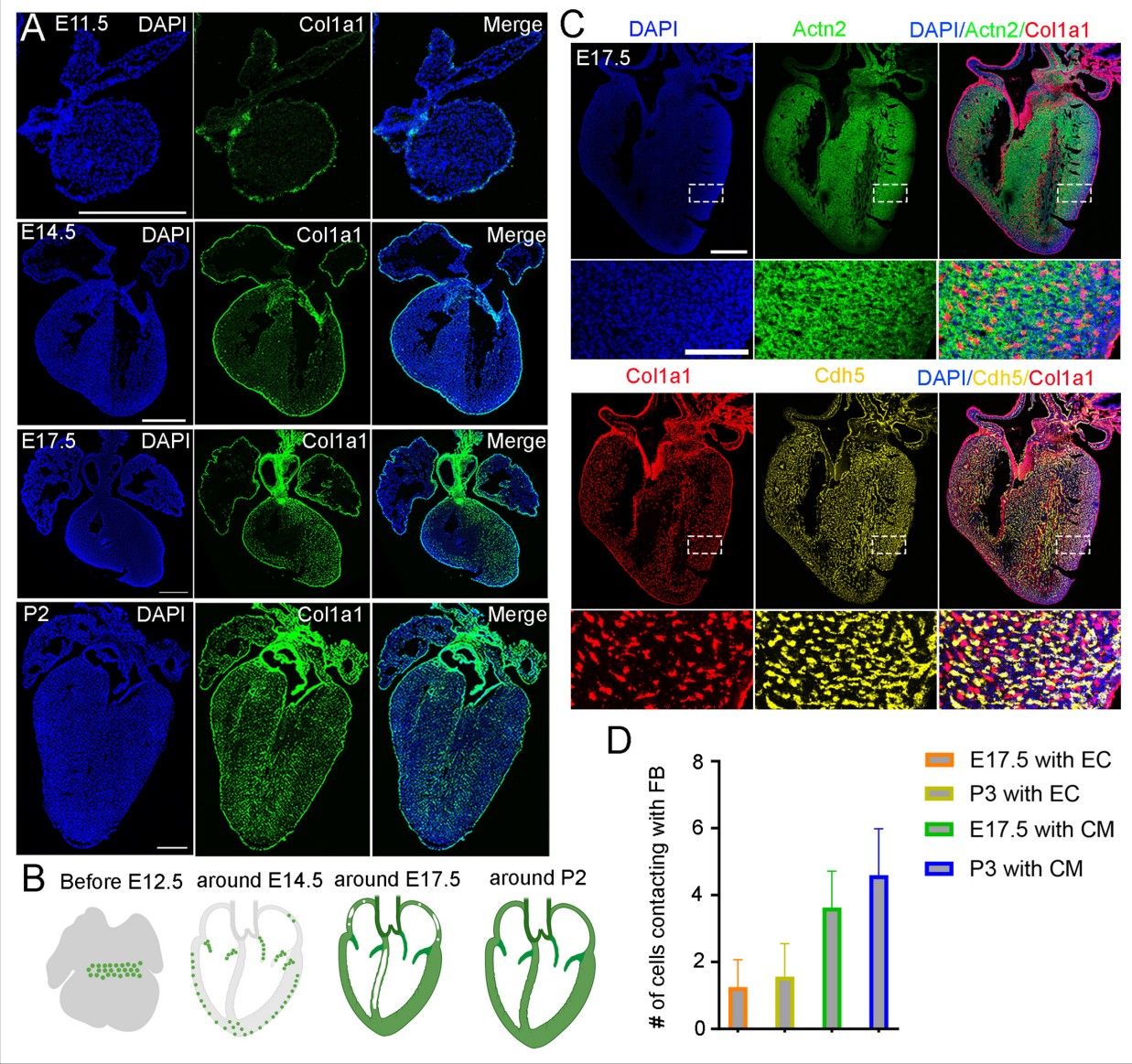

**Figure 1.** Anatomical location of cardiac fibroblasts (FBs) and their spatial relationship with cardiomyocytes and endothelial cells at different stages. (**A**) RNA staining of *Col1a1* revealed the spatial pattern of cardiac FBs at different developmental stages. Scale bar = 500 μm. (**B**) The development of cardiac FBs can be grouped into four phases. (**C**) RNA staining analysis of *Col1a1*, *Actn2*, and *Cd31* revealed the spatial proximity of FB, CM, and EC in E17.5 hearts. (**D**) Quantification of the number of CMs and ECs that contact with each FB (n=100). Scale bar = 500 μm and 100 μm in the whole heart sections and enlarged sections, respectively.

The online version of this article includes the following figure supplement(s) for figure 1:

**Figure supplement 1.** The anatomical patterns of fibroblasts at different stages.

**Figure supplement 2.** The anatomical patterns of fibroblasts, cardiomyocytes, and endothelial cells.

**Figure supplement 3.** ScRNA-seq identified distinct fibroblast populations.

---

the valve interstitial cells highly expressed *Col1a1* (*Figure 1A*, *Figure 1—figure supplement 1*). In the ventricles, a small proportion of *Col1a1*-positive cells were observed adjacent to the epicardium. In contrast, few *Col1a1* signals were found in the atrium at this stage (*Figure 1A*). At E15.5 and 16.5, both ventricles were filled with *Col1a1*-positive cells, but the middle of the ventricular septum remained devoid of such cells (*Figure 1A*, *Figure 1—figure supplement 1*). FBs were also found in the atrium, but the signal at the tips of both atrial chambers was sparser (*Figure 1A*, *Figure 1—figure supplement 1*). The distribution of FBs at E17.5 and E18.5 was highly similar to that observed at E16.5

(*Figure 1A*, *Figure 1—figure supplement 1*). Furthermore, we analyzed the *Col1a1* signal in P0, P2, and P3 hearts and found dense *Col1a1* expression at the valves and in all four chambers (*Figure 1A*, *Figure 1—figure supplement 1*). In summary, this analysis revealed the spatial location of cardiac FBs along the developmental stages. Specifically, we found that the FBs developed first in the AVC and valve, then at the ventricular wall and atrial, and last at the ventricular septum (*Figure 1B*).

Next, to analyze the spatial relationship between FB and other cardiac cell types, such as CM and EC, we performed RNA staining for their marker genes *Col1a1*, *Actn2*, and *Cdh5*. Consistent with the previously described pattern, we observed that FB were mainly enriched in valves and began to develop at the edges of ventricular chambers at E13.5. We also found that FB were intermingled with CM populations and spatially adjacent to ECs (*Figure 1—figure supplement 2A*). Next, we stained the three lineage markers on E17.5 and P3 heart sections. We observed consistent results as E13.5 (*Figure 1C*, *Figure 1—figure supplement 2B*). Given that FB starts emerging at E13.5, we primarily quantified the contacts at E17.5 and P3. The quantification results showed that each FB was in contact with approximately one Vas_EC and four CMs at both stages (*Figure 1C and D*). These results suggest that cardiac FBs are physically close to CMs and ECs, likely facilitating signal communications that impact their development.

## Identification of heterogeneity in cardiac FBs through the analysis of scRNA-seq data

Through the analysis of scRNA-seq data at 18 stages of developing hearts from CD1 mice, we investigated the expression of *Col1a1* and observed high expression in epicardial cells and FBs (*Figure 1—figure supplement 3A*). This is consistent with the findings from RNA staining (*Figure 1A*). Subsequently, we reanalyzed the FBs and identified four distinct populations comprising 13 clusters (*Figure 1—figure supplement 3D*). Gene expression analysis revealed that all cells in these populations exhibited high expression of *Col1a1* (*Figure 1—figure supplement 3A*). Moreover, through differential gene expression analysis between the populations, we discovered that one group of cells highly expressed *Hapln1* (*Figure 1—figure supplement 3B, D, E*), a marker gene associated with valves. This indicates that these cells are valve interstitial cells. We also found another group of cells expressing *Cdh5* and *Tie1* (*Figure 1—figure supplement 3B, D, E*), genes associated with the endothelial cell lineage. This suggests that these cells may represent FBs derived from EDCs. Additionally, we identified a third group of FBs expressing *Sox10* and *Phox2b* (*Figure 1—figure supplement 3B, D, E*), genes associated with the neural crest cell lineage. This suggests that these cells may originate from neural crest cells. Finally, the rest of the cells formed a main population that likely represents epicardial cell-derived FBs in the chambers, as they expressed *Tcf21* and other marker genes (*Figure 1—figure supplement 3D, E*). Interestingly, further analysis of the cell cycle phases in these FBs revealed that each population consisted of cells in all three phases (*Figure 1—figure supplement 3C*), indicating active proliferation during the early stages.

Next, to understand the anatomical location of these FB populations, we performed RNA in situ hybridizations on postnatal heart sections. Firstly, we stained *Hapln1* together with *Dcn*, another Pan-FB marker gene. We found that *Hapln1* was highly expressed in valves, and most of its signal overlapped with *Dcn* (*Figure 1—figure supplement 3Fi*), suggesting its expression in valve interstitial cells (FB-like cells in valves). We also stained *Cdh5* with *Col1a1* and observed a small cluster of cells with the expression of both genes, mostly located at the boundary of large vessels and ventricular chambers (*Figure 1—figure supplement 3Fii*). Finally, we co-stained *Sox10* and *Col1a1* expression (*Figure 1—figure supplement 3Fiii*). As previously reported (*Ali et al., 2014*), we identified double-positive cells in the outflow tract-derived large vessels (*Figure 1—figure supplement 3Fiii_1*). However, interestingly, we also observed their presence in coronary vessels (*Figure 1—figure supplement 3Fiii_2*), suggesting that these cells may have a broader function than previously thought. In summary, through the analysis of scRNA-seq data and in situ hybridizations, we identified different populations of cardiac FBs, which differentially expressed lineage genes likely from their precursor cells.

## Molecular analysis of the signaling communications between FBs and the other cardiac cell types

To understand the interactions between FB and other cardiac cell types, we analyzed scRNA-seq data from developing CD1 hearts at 18 stages ranging from E9.5 to P9 (*Feng et al., 2022*). By quantifying the number and strength of these interactions, we found that each cell type, except blood cells, had a similar number of interactions. However, in terms of interaction strength, FB exhibited the highest values compared to those of other cell types (*Figure 2A*, *Supplementary file 2*). Next, we analyzed the signaling pathways secreted from FB to ventricular cardiomyocytes (Ven_CM) and Vas_EC. Interestingly, we found that the collagen pathway was the predominant pathway in both FB-VenCM and FB-VasEC interactions (*Figure 2B*). In addition to collagen, we identified other signaling molecules such as laminin, Ptn, MK, Fn1, and Thbs, which are mostly extracellular matrix (ECM) proteins, as being secreted by FB to function in Ven_CM and Vas_EC (*Figure 2B*). Within the collagen pathway, we further analyzed the detailed ligand-receptor interactions and found more interaction pairs in FB-VasEC than in FB-VenCM. Some interactions were shared between the two cell types, but most were distinct (*Figure 2C*). Consistently, all the shared collagen types have been previously studied and found to play a crucial role in heart development, primarily by regulating AVC and vasculature development (*Kruithof et al., 2007*; *Lincoln et al., 2004*; *Lincoln et al., 2006*; *van der Kooi et al., 2006*; *Lockhart et al., 2011*).

To better understand the role of ECM genes in FB_VasEC and FB_VenCM interactions, we analyzed their expressions (GO number, 0031012) in the chamber-derived fibroblasts (main_fb) using the CD1 and C57BL/6 scRNA-seq datasets from 18 stages (*Feng et al., 2022*; *Figure 2—figure supplement 1A–F*). Interestingly, we identified five groups of ECM genes that displayed stage or chamber-specific expression in both datasets (*Figure 2—figure supplements 2–4*, *Supplementary file 3* and *Supplementary file 4*). Specifically, we found one group of genes (G1) that was highly expressed at stages before E14.5. This gene group includes signaling molecules such as *Slit2* and *Wnt5a*, and collagen genes such as *Col13a1* and *Col26a1* (*Figure 2—figure supplements 2–4*). We found the second group of genes (G2) was mainly expressed in the LV and RV from E14.5 onwards with a significant increase at the neonatal stage. This gene group includes collagen genes such as *Col5a1*, *Col6a1*, and *Col6a2*, and metalloproteinase genes such as *Adamts13* and *Adamts14*. Additionally, we identified that the third group of genes was highly expressed in the LA and RA at most stages, which includes *Col12a1*, *Adamts1*, and *Adamts4*. Interestingly, *Fn1* also exists in this group, suggesting that it has an important function in atrial FB development. Furthermore, we found the fourth group of genes was preferentially expressed in the other three chambers than the RA and was mainly expressed at late neonatal stages after P5 in certain chambers. This group of genes includes *Col4a4*, *Col4a5*, *Col4a6*, and *Ntn4*. Finally, we identified the fifth group of genes displayed universal expression in all four chambers at most stages. This gene group includes *Col1a1*, *Col1a2*, and *Postn*, which are the ubiquitously expressed extracellular matrix genes (*Figure 2—figure supplements 2–4*). In summary, we identified five groups of ECM genes displaying stage- and chamber-specific expression patterns in FB and observed that each gene group includes different collagen pathway members. Additionally, we analyzed the ligand-receptor interactions between FB and Ven_CM or Vas_EC at each stage. We found collagen pathway interactions were also enriched at most analyzed stages (*Figure 2—figure supplement 5A, B*, *Supplementary file 5* and *Supplementary file 6*). These results together suggested the importance of the collagen pathway in mediating the FB function in heart development.

Furthermore, we studied collagen deposition dynamics in developing hearts. We used collagen hybridizing peptide (CHP) to stain the total collagen after denaturing it via an antigen retrieval process. Consistent with *Col1a1* expression, we observed strong collagen signals in the epicardium at all analyzed stages (*Figure 2D*). Interestingly, at E11.5, we observed collagen in both AVC and ventricular chambers. Given that FBs in the chambers have not developed yet at this stage, the observed collagen is likely derived from other cell types, such as endocardial endothelial cells. At E14.5, we observed a strong collagen signal in the ventricular free walls, while the signal in the ventricular septum and atrium was relatively weaker. At E16.5, the collagen signal became brighter and denser throughout the heart, including the ventricular free wall, septum, and atrium. This distribution was slightly broader compared to the fibroblast distribution at the same stage (*Figure 1A*). By P3, the collagen signal had further increased and formed stripe patterns (*Figure 2D*). In summary, we found that the progression of collagen accumulation largely coincided with the development of FBs.

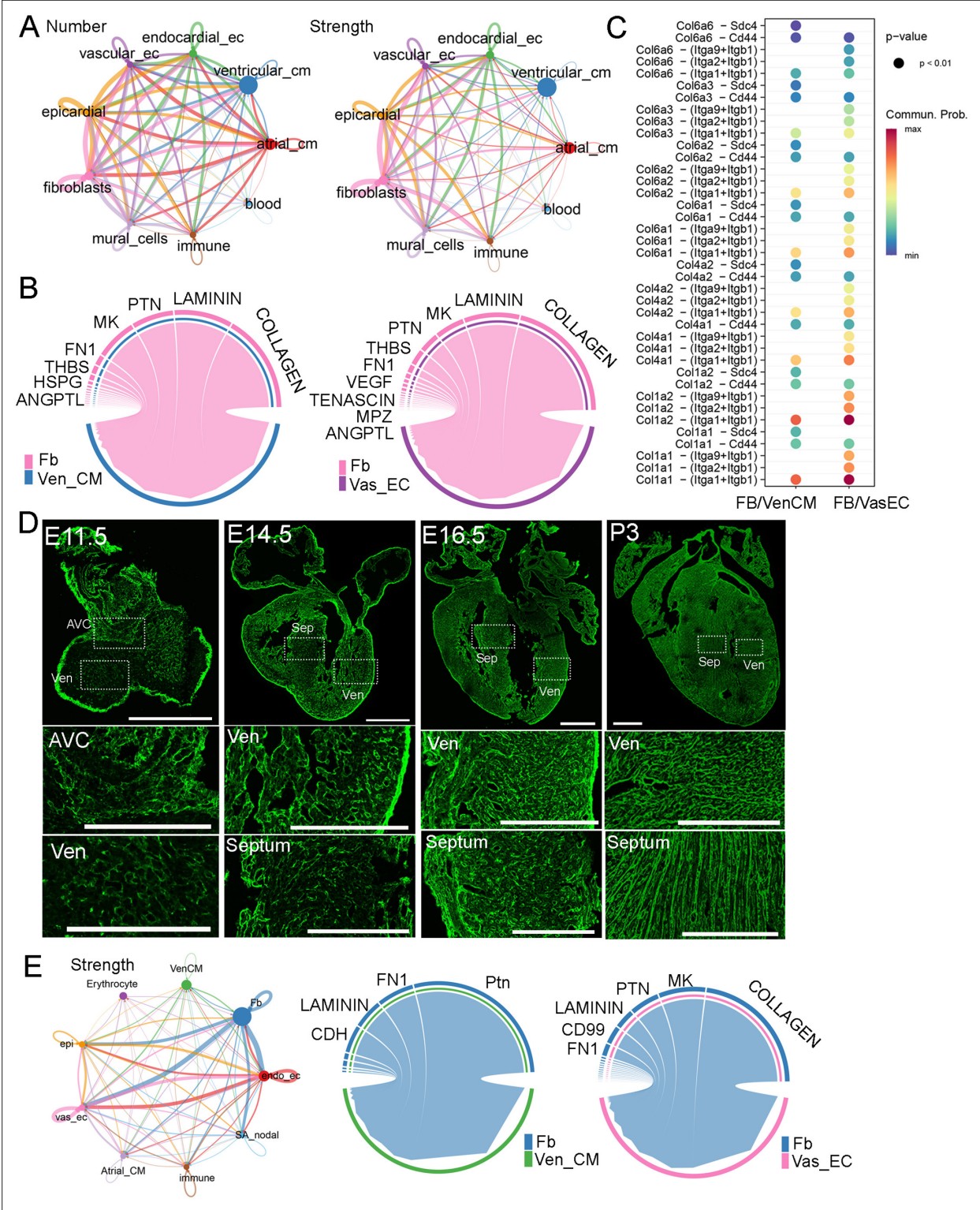

**Figure 2.** Molecular analysis of fibroblast (FB) communications with other cardiac cell types in developing hearts. (**A**) The number and strength of interactions among different cardiac cell types identified from scRNA-seq data. (**B**) The top signaling pathways derived from FB that impact Ven_CM and Vas_EC development. (**C**) The detailed ligand-receptor interactions in the collagen pathway between FB and Ven_CM or Vas_EC. (**D**) Collagen was visualized using CHP staining in mouse hearts at E11.5, E14.5, E16.5, and P3. Scale bar = 500 μm and 150 μm in the whole heart sections and enlarged sections, respectively. (**E**) The signaling interactions and top pathways between FB and Ven_CM or Vas_EC in primary human hearts.

The online version of this article includes the following figure supplement(s) for figure 2:

*Figure 2 continued on next page*

*Figure 2 continued*

**Figure supplement 1.** Heterogeneity analysis of cardiac fibroblasts.

**Figure supplement 2.** The expression pattern of extracellular matrix genes in the main population of cardiac fibroblasts.

**Figure supplement 3.** The expression pattern of group G5 genes in fibroblasts.

**Figure supplement 4.** The expression pattern analysis of extracellular matrix genes in C57BL/6 FBs.

**Figure supplement 5.** The molecular interactions between FBs and other cell types.

Lastly, we analyzed the signaling interactions among human cardiac cells using a human embryonic heart scRNA-seq dataset (*Asp et al., 2019*). Consistent with the mouse analysis results, we identified strong interactions between FB and other cardiac cell types (*Figure 2E*). Additionally, we found collagen to be the most enriched pathway in FB-VasEC interactions. However, interestingly, we did not observe the collagen pathway among the top enriched pathways in FB-VenCM interactions (*Figure 2E*, *Figure 2—figure supplement 5C*). This observation was confirmed in two other human embryonic heart scRNA-seq datasets (*Suryawanshi et al., 2020*; *Cui et al., 2019*). These results suggest a partially similar mechanism underlying FB regulation of heart development in both mice and humans.

## Functional analysis of FBs in embryonic heart development

To identify the right transgenic mice labeling the developing cardiac FBs, we analyzed the expression of two well-known FB marker genes, *Postn* and *Pdgfra*, in the CD1 scRNA-seq data. We observed that *Postn* was highly expressed in FBs and mural cells. Among FBs, *Postn* was expressed in all subpopulations (*Figure 3—figure supplement 1Ai*). On the other hand, *Pdgfra* was specifically expressed in FBs, although its expression was relatively lower than *Postn*, especially in postnatal staged atrial cells (*Figure 3—figure supplement 1Aii*). To evaluate their efficiency in labeling FBs, we bred *Postn-CreER* mice with *Rosa-mTmG* reporter mice and administered tamoxifen (200 μg/g) to pregnant mice at E13.5 (*Figure 3—figure supplement 1Bi*). We collected the hearts at E17.5 and observed only a small proportion of GFP-positive cells in the valves and septum (*Figure 3—figure supplement 1Bii*), suggesting a low labeling efficiency from the *Postn-CreER* mice. In contrast, we conducted the same experiments using *Pdgfra-CreER* mice and observed a significant proportion of GFP-positive cells in both the valves and heart chambers (*Figure 3A*). These results demonstrate the efficient labeling capability of the *Pdgfra-CreER; Rosa-mTmG* mice.

To study the function of FBs in heart development, we utilized a genetically encoded DTA system to ablate them, which involves the encoding of a toxin gene that induces apoptosis and allows for the ablation of target cells. Initially, we crossed floxed *Rosa-DTA* mice with *Pdgfra-CreER* mice and administered tamoxifen to pregnant female mice at E10.5. After 3 days, we harvested the embryos and observed that all the embryos with ablated FBs (*Pdgfra-CreER+/-; Rosa-DTA+/-*) had perished, while the control embryos remained unaffected. This outcome indicates an essential role for FBs in early-stage embryo development.

Subsequently, we administered tamoxifen to the female mice from the breeding pair at E13.5 and collected the embryonic hearts at E16.5 (*Figure 3B*). Our findings revealed that the ablated embryos exhibited smaller sizes compared to the control embryos, although the size of the ablated hearts did not significantly differ from that of the control hearts (*Figure 3B*). Additionally, we conducted TUNEL analysis to assess cell death in both control and ablated hearts and observed a significantly higher number of TUNEL-positive dots in the valve and chamber regions of the ablated hearts compared to the control hearts, confirming the successful ablation of FBs in the experimental hearts (*Figure 3—figure supplement 2A*, B). Furthermore, we examined the anatomical structure of the hearts but did not identify obvious defects in the ablated groups compared to control groups. However, we observed a reduction in endothelial cell density in the ablated hearts, as indicated by the CD31 staining signal (*Figure 3C*). Additionally, we measured the thickness of the compact and trabecular myocardium (*Figure 3—figure supplement 2C*). Our observations revealed that, compared to the control hearts, the ablated hearts exhibited a thinner left ventricular (LV) compact myocardium, while the thickness of the LV trabecular myocardium remained similar. Furthermore, we noted an elevated ratio of LV trabecular to compact myocardium in the ablated hearts compared to the control hearts (*Figure 3D*). However, no similar differences were observed in the right ventricle (RV). Finally, we examined cell

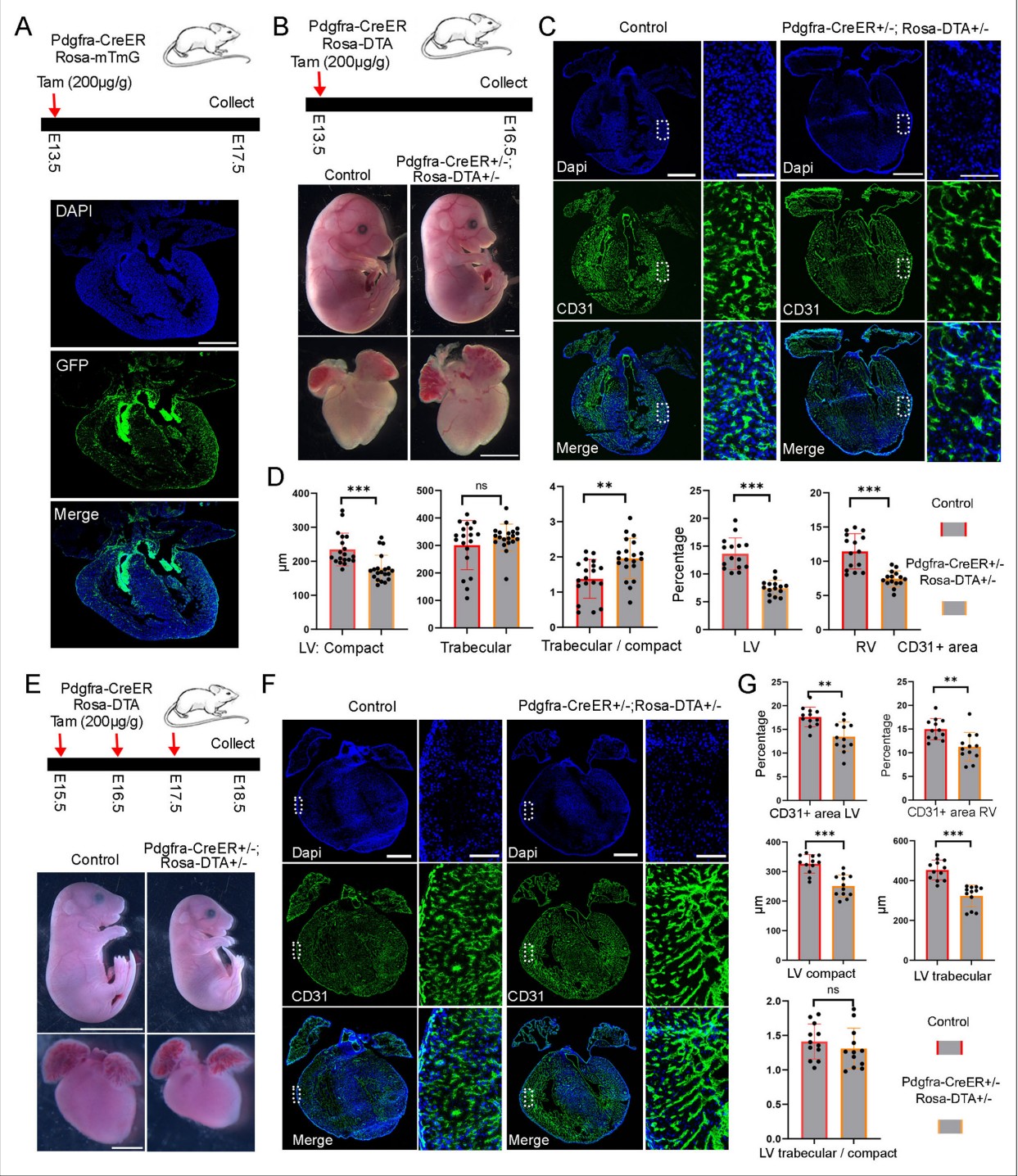

**Figure 3.** Functional analysis of FB in embryonic heart development. (**A**) Diagram of the lineage-tracing experiments and the tracing results. Scale bar = 500 μm. (**B**) The ablated embryos treated with tamoxifen at E13.5 were smaller than control embryos, but the ablated hearts were not obviously different from the control hearts. Scale bar = 1 mm. (**C**) Representative CD31 staining images in control and ablated hearts. The zoomed-in images showed a reduction of CD31-positive area in the ablated hearts compared to controls. (**D**) Quantification of the compact and trabecular myocardium thickness (19 sections from 7 ablated hearts and 20 sections from 7 control hearts), and CD31-positive areas in control and ablated hearts (15 sections from 5 ablated and 15 sections from 5 control hearts). The littermate control mice include both double-negative and Pdgfra-CreER +/−mice. (**E**) The ablated mice and hearts treated with tamoxifen at E15.5 were smaller than controls. (**F**) CD31 staining analysis of control and ablated hearts. (**G**) Quantification of the compact and trabecular myocardium thickness, and CD31-positive areas in control and ablated hearts (12 sections from 4 ablated hearts and 12 sections from 4 control hearts.). The littermate control mice include both double-negative and Pdgfra-CreER +/−mice. Scale bar = 500 μm in the whole heart sections and 100 μm in the zoomed-in images. * represents p<0.05; ** represents p<0.01.

*Figure 3 continued on next page*

*Figure 3 continued*

The online version of this article includes the following figure supplement(s) for figure 3:

**Figure supplement 1.** Comparative analysis of the expression patterns and lineage tracing results between the genes Postn and Pdgfra.

**Figure supplement 2.** Comparative analysis of cell death and myocardial thickness in control and ablated hearts.

**Figure supplement 3.** Comparative analysis of valve development in control and fibroblast-ablated hearts at E18.5.

**Figure supplement 4.** Quantification of the defects in control and ablated hearts with one dose of tamoxifen treatment at E15.5.

proliferation post-ablation using pHH3 staining. We observed a reduction in pHH3-positive cells in the ventricular region, but not in the atrial region, of the ablated hearts compared to the control hearts (*Figure 3—figure supplement 2B*). These results suggest that cell proliferation in the ventricle was impaired following FB ablation.

Next, we performed ablation at late embryonic stages by treating the pregnant mice with tamoxifen at E15.5 and harvested the embryos at E18.5 (*Figure 3—figure supplement 4A*). We did not observe obvious differences in the size of the embryos and hearts between the ablated group and the control group (*Figure 3—figure supplement 4B*). However, the CD31 staining of heart sections revealed large holes in the ablated atrium but not in the ablated ventricles or control hearts (*Figure 3—figure supplement 4C*). Additionally, quantification of the compact and trabecular myocardium thickness in the LV and RV revealed a thinner LV compact myocardium (*Figure 3—figure supplement 4D*). We also calculated the ratio of trabecular to compact myocardium thickness in the LV and RV and found that the ratio was increased in the RV but not in the LV of the ablated hearts compared to control hearts (*Figure 3—figure supplement 4E*). These results indicated that both the LV and RV of the ablated hearts have subtle defects. Moreover, we analyzed cell proliferation by staining pHH3 and found no significant differences between the ablated and control ventricles (*Figure 3—figure supplement 4F*).

Given that the animals did not show obvious morphological changes after one dose of tamoxifen treatment, we increased the tamoxifen treatment to three consecutive days, ranging from E15.5 to E17.5 (*Figure 3E*). We harvested the embryos at E18.5 and found that both the ablated embryos and their hearts were notably smaller in size compared to the controls (*Figure 3E*). Next, to confirm the ablation efficiency, we analyzed *Col1a1* expression with RNA staining and found that the ablated hearts lost almost all of their *Col1a1* signal in the four chambers (*Figure 3—figure supplement 4G*), indicating a high ablation efficiency with three doses of tamoxifen treatments. We further analyzed their valve structures and observed that the valves in the ablated hearts were not severely affected (*Figure 3—figure supplement 3A*). Moreover, by conducting a staining analysis of CD31 in the control and ablated heart sections, we discovered that the ablated hearts exhibited a significantly lower endothelial cell density (*Figure 3F and G*). Further quantification of the LV compact and trabecular myocardium thickness showed a significant reduction in the ablated hearts compared to control hearts (*Figure 3G*). However, interestingly, the ratio of trabecular to compact myocardium thickness in the LV did not differ between the control and ablated hearts, indicating a proportional reduction in both compact and trabecular myocardium thickness in the ablated hearts. Finally, we found that the ablated ventricles had less pHH3 signal than the control ventricles (*Figure 3—figure supplement 4F*), indicating impaired cell proliferation in the ventricles following FB ablation. These results together indicated that FB plays an important role in embryonic heart development at all the analyzed stages.

## Molecular analysis of the defects in the ablated embryonic hearts

Next, we performed scRNA-seq to analyze the defects in the ablated embryonic hearts. We utilized MULTI-seq to pool the control and ablated hearts from E16.5 and E18.5 (*Figure 4A*). After filtering out low-quality cells based on our QC standards, we grouped the single cells from two experimental replicates using unsupervised methods and found a high degree of overlap (*Figure 4B*, *Figure 4—figure supplement 1A*). We then identified cell types and cell cycle phases within the single cells. In total, we identified eight cell types: Atrial_CM, Ven_CM, Vas_EC, Endo_EC, Mural_cell, FB, Epicardial cell, and Immune cell, along with three cell cycle phases (G1, S, G2M) in each cell type (*Figure 4C*). Cells from both control and ablated conditions at E16.5 and E18.5 were present in each cell type. We further quantified the percentages of each cell type across conditions and found that the percentage of FB was significantly reduced in the ablated hearts at both stages, indicating effective ablation. Interestingly, we also observed a reduction in mural cells in the ablated hearts at both stages (*Figure 4D*).

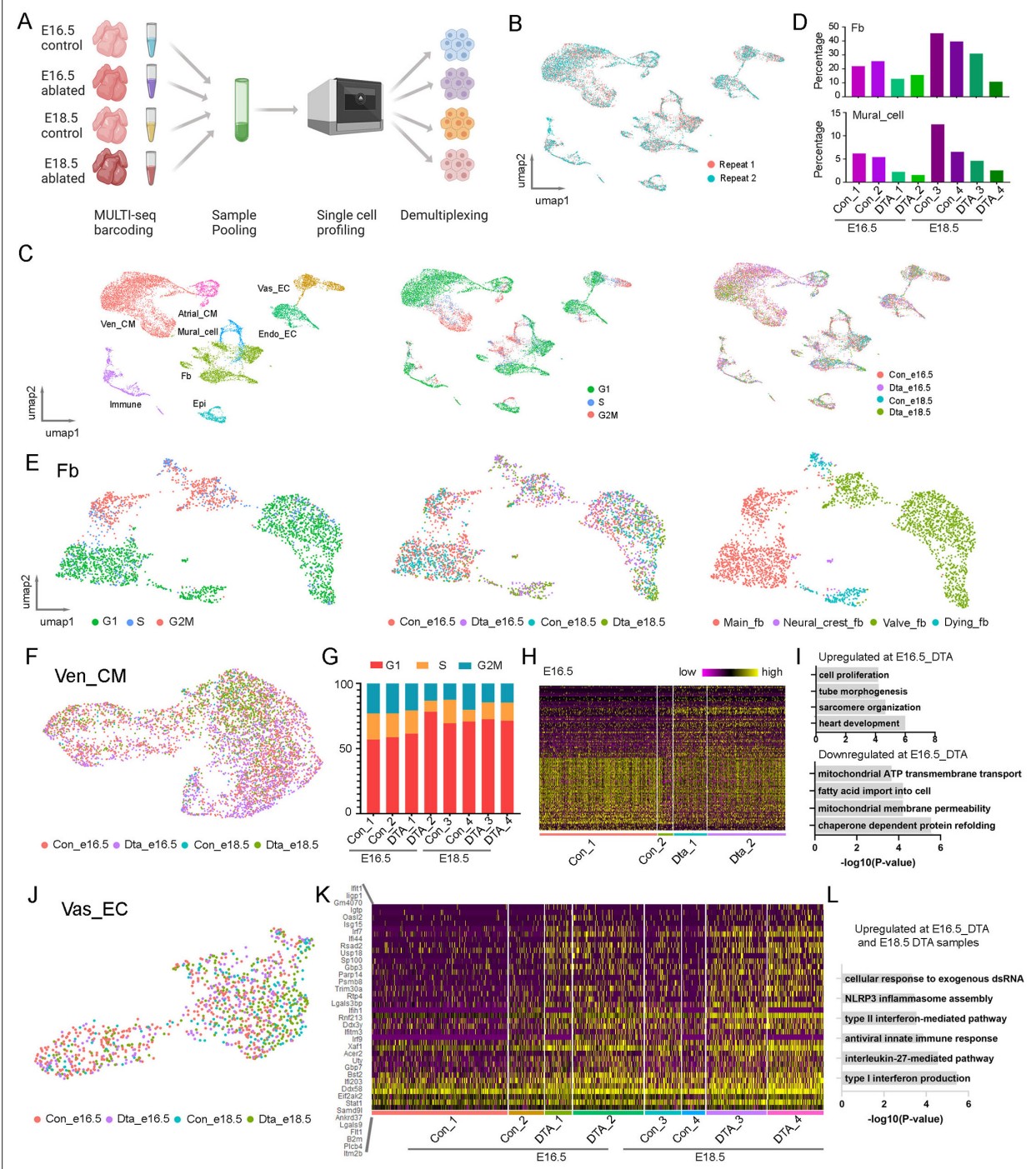

**Figure 4.** Single-cell analysis of the embryonic heart defects after fibroblast (FB) ablation. (**A**) Diagram of the MULTI-seq experiments to profile control and ablated hearts at two developmental stages. (**B**) The scRNA-seq data from two replicates are highly consistent. (**C**) UMAP plots of scRNA-seq data labeled by cell type, cell cycle phase, and genotype. (**D**) Quantification of the FB and mural cell percentages at each condition. (**E**) Detailed analysis of the FB population revealed four groups, including a dying FB subpopulation. (**F**) UMAP plot of Ven_CMs from different conditions. (**G**) Quantification of the different cell cycle phased Ven_CMs under each condition. (**H, I**) Heatmap and pathway enrichment of genes that are differentially expressed in control and ablated Ven_CMs at E16.5. (**J**) UMAP plot of Vas_EC labeled by conditions. (**K, L**) Heatmap and pathway enrichment of genes that were upregulated in ablated Vas_ECs compared to control Vas_ECs. The littermate control mice at both stages were Pdgfra-CreER+/−.

The online version of this article includes the following figure supplement(s) for figure 4:

**Figure supplement 1.** Plots of QC results, expression of Fb sublineage genes, and annotation of Ven_CM and Vas_EC with cell cycle phases.

Next, we selected FB for further study of its cellular heterogeneity. By analyzing the expression of cardiac lineage genes, cell cycle phases, and sample sources, we identified four FB subpopulations (*Figure 4E*). Specifically, we identified a group of valve fibroblasts (valve-fb) expressing valve mesenchymal cell genes such as *Hapln1*, a group of neural crest-derived fibroblasts (Neural_crest_fb) expressing *Sox10*, and a group of chamber-derived main fibroblasts (main_fb) expressing *Wt1* and *Tcf21* (*Figure 4—figure supplement 1B*). All these FB groups contained cells from both control and ablated samples. In contrast, we identified another group of cells exclusively derived from the ablated samples, which expressed cell death-related genes (*Figure 4—figure supplement 1C*). We named this group dying fibroblasts (dying_fb).

In addition to FB, we analyzed Ven_CM and found that cells from control and ablated hearts were highly overlapped on the UMAP plot (*Figure 4F*). We quantified the proportions of cells at each cell cycle phase and found that the ablated samples had slightly higher percentages of cells in the G1 phase compared to control samples (*Figure 4G*, *Figure 4—figure supplement 1D*). We then performed differential gene expression analysis of control and ablated samples at E16.5 and identified a set of differentially expressed genes (*Figure 4H*, *Supplementary file 7*, *Supplementary file 8*). Gene ontology analysis of the upregulated genes in the ablated CMs showed enrichment in various pathways, including heart development, sarcomere organization, heart tube morphogenesis, and cell proliferation. In contrast, the downregulated genes were enriched in pathways related to chaperone-dependent protein folding, mitochondrial membrane permeability, fatty acid import, and mitochondrial ATP transmembrane transport (*Figure 4I*). These results indicated that ablated Ven_CMs upregulated genes associated with heart development and cell proliferation while down-regulating genes related to cell maturation, including those involved in mitochondrial function and cell metabolism.

Furthermore, we selected Vas_EC for further analysis. We did not observe distinct populations between control and ablated samples on the UMAP plot (*Figure 4J*, *Figure 4—figure supplement 1E*). However, through further differential expression analysis, we identified a group of genes with upregulated expression in the ablated samples (*Figure 4K*, *Supplementary file 9*). Gene ontology analysis of these genes revealed that they were primarily involved in immune response pathways, including type I and II interferon production, interleukin-27-mediated pathways, NLRP3 inflammasome assembly, and cellular responses to exogenous dsRNA (*Figure 4L*). These results suggest that Vas_EC exhibits a strong immune response following FB ablation.

## The signaling pathway defects in the ablated embryonic hearts

Next, we studied the signaling defects in the ablated hearts. Through the analysis of signaling pathways between main_fb and Vas_EC or Ven_CM, we found that the collagen pathway consistently ranked at the top in both control and ablated samples (*Figure 5A*). We then stained for collagen levels using collagen hybridizing peptide and observed that collagen deposition was dramatically reduced in both the septum and ventricular wall of the ablated hearts compared to the control hearts (*Figure 5B*).

Moreover, by analyzing the scRNA-seq data, we predicted the signaling ligands that may regulate the differentially expressed genes in control and ablated Ven_CM. Interestingly, the collagen pathway, which has four different collagen ligands on the list, emerged as a major regulator of these genes (*Figure 5C*). In addition to collagens, we also identified other ligands, such as TGF-beta, IGF, and BMP, which were derived from FB and regulate the expression of differentially expressed genes in control and ablated Ven_CMs (*Figure 5C*). We also predicted the receptors in Ven_CM for each ligand (*Figure 5—figure supplement 1A*). Furthermore, we predicted the ligands in FB that regulate the differentially expressed genes in Vas_EC and identified several signaling molecules. These included the DTA pathway-related signal Hbegf, the inflammation-associated signal Il33, Wnt ligands (Wnt11, Wnt4, and Wnt5b), and collagen Col4a6 (*Figure 5D*, *Figure 5—figure supplement 1B*).

Finally, we predicted the signals from the dying_fb cells. Compared to normal main_fb and valve_like_fb, the dying_fb downregulated certain signaling pathways, such as the IGF pathway, EphA pathway, Tenascin pathway, and collagen pathway (*Figure 5E*, *Figure 5—figure supplement 2A*). Interestingly, they also upregulated several signaling pathways, including the EGF pathway, BMP pathway, VEGF pathway, and FGF pathway (*Figure 5E*, *Figure 5—figure supplement 2B*). Understanding the role of these altered signaling pathways from dying_fb in contributing to heart defects

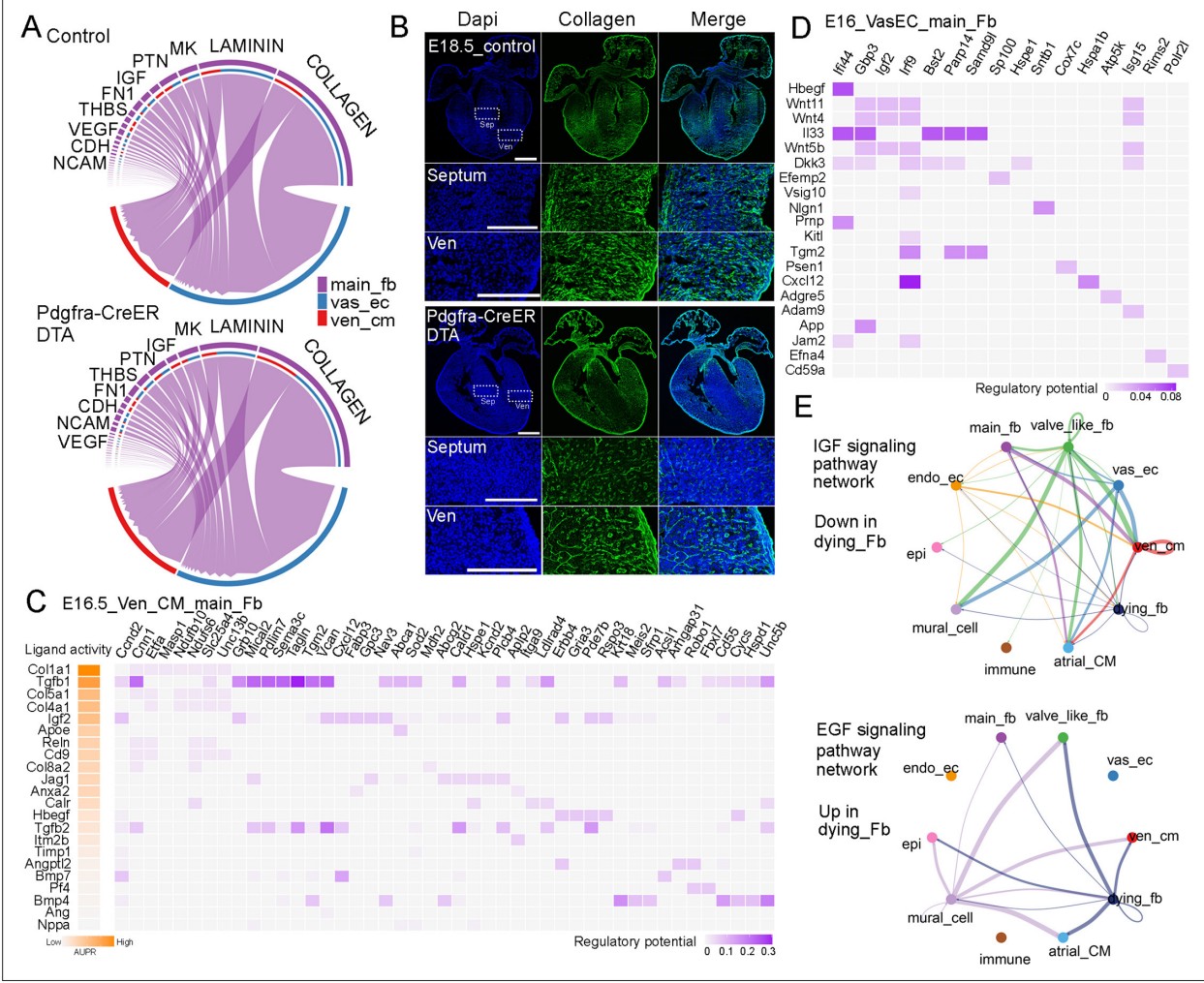

**Figure 5.** Identification of signaling defects in ablated hearts. (**A**) The top signaling pathways between FB and Ven_CM or Vas_EC in control and ablated hearts. (**B**) CHP staining revealed a significant reduction in collagen deposition in ablated hearts compared to control hearts. (**C, D**) Regulatory analysis predicted the ligands that regulate the genes differentially expressed in control and ablated Ven_CMs and Vas_EC. (**E**) Representative signaling pathways that were downregulated or upregulated in the dying FBs.

The online version of this article includes the following figure supplement(s) for figure 5:

**Figure supplement 1.** The predicted ligand-receptor interaction potentials between main_fb and Ven_CM or Vas_EC.

**Figure supplement 2.** The signaling pathways that were found to reduce or increase their strengths in the dying_Fb compared to the main_fb and valve_like_fb.

will be an interesting area for future research. Additionally, collagen and other ECM proteins are essential structural components. Their roles as structural proteins in heart development are critical and contribute to the heart phenotype observed in the ablated hearts.

## Functional analysis of FBs at neonatal stages

Next, we assessed the FB function at neonatal stage. We treated the mice with tamoxifen at P1 and harvested them at P4 (*Figure 6—figure supplement 1A*). We did not observe any obvious differences between the ablated and control hearts (*Figure 6—figure supplement 1B*), consistent with a recent study that used the same strategy to ablate FBs during a similar time period (*Kuwabara et al., 2022b*). However, CD31 staining of heart sections revealed defects in the atrium, which had a thinner and segmented atrial wall (*Figure 6—figure supplement 1C*). We further quantified the compact and trabecular myocardium in both the LV and RV but did not observe differences between control and ablated hearts (*Figure 6—figure supplement 1D*). We also did not find differences in the ratio

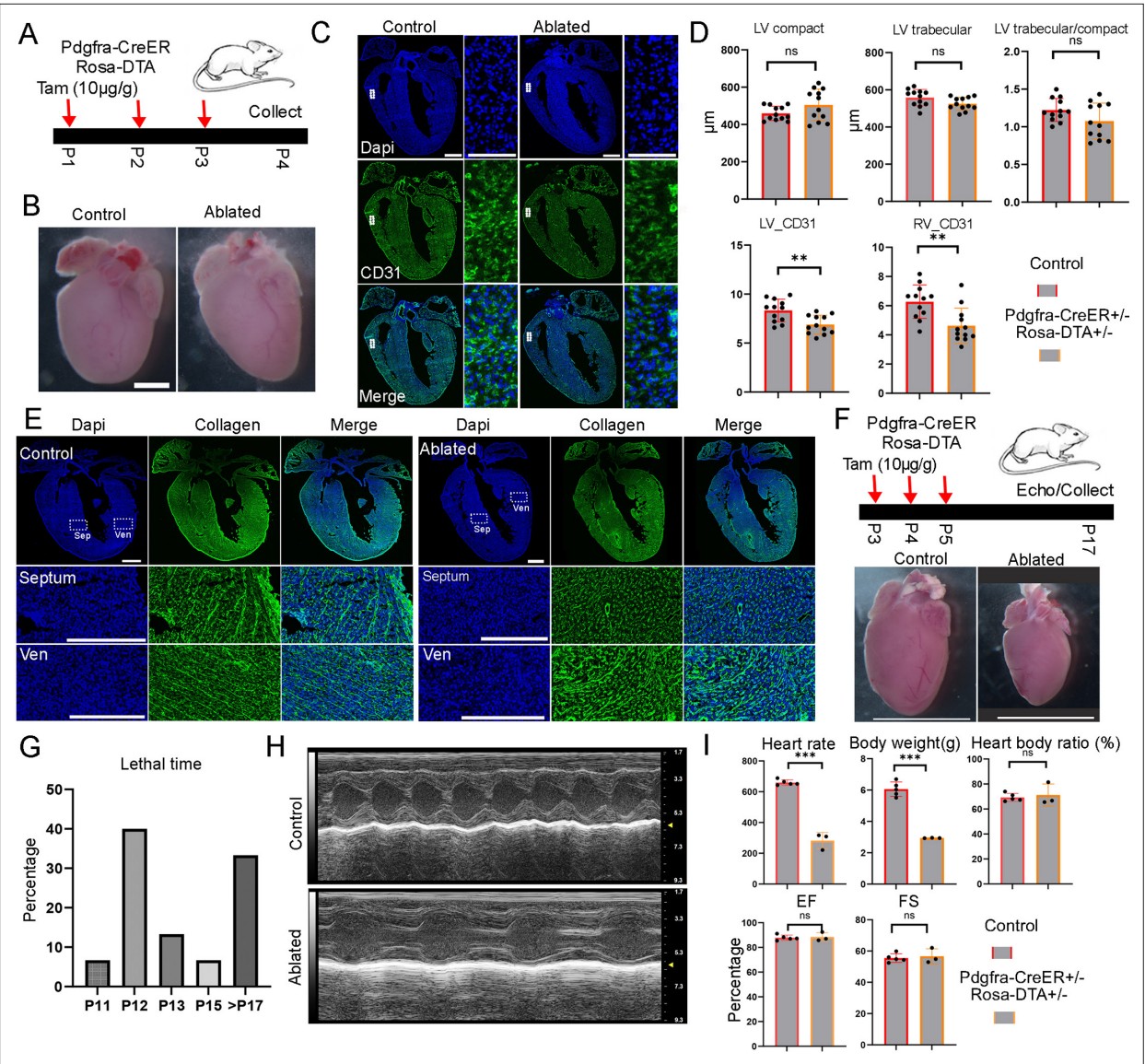

**Figure 6.** Short-term and long-term functional analysis of fibroblasts at neonatal stage. (**A**) Diagram of the experiment to ablate fibroblasts with three doses of tamoxifen treatment from P1 to P3. Scale bar = 1 cm. (**B**) No obvious size difference was observed between control and ablated hearts. (**C**) Representative images of control and ablated hearts stained with CD31. (**D**) Quantification of the compact and trabecular myocardium thickness and CD31-positive areas in ventricular at control and ablated hearts (12 sections from 4 ablated hearts and 12 sections from 4 control hearts). (**E**) Collagen accumulation in control and ablated hearts at P4. Collagen was visualized using CHP staining. The littermate control mice were Pdgfra-CreER+/−. Scale bar = 500 μm and 150 μm in the whole heart sections and enlarged sections, respectively. (**F**) Tamoxifen was given from P3 to P5 to ablate the fibroblasts, and hearts were collected at P17 for analysis. The littermate control mice were Pdgfra-CreER+/−. (**G**) The percentage of mice that died at different days. (**H**) Representative echo images of the control and ablated hearts. (**I**) Quantification of the heart rate, body weight, and heart function in control and ablated mice at P17. * represents p<0.05; ** represents p<0.01; *** represents p<0.001.

The online version of this article includes the following figure supplement(s) for figure 6:

**Figure supplement 1.** Quantification of the defects in control and ablated hearts at neonatal stage with one dose of tamoxifen treatment.

**Figure supplement 2.** Collagen analysis at P18 hearts.

**Figure supplement 3.** Effects of fibroblast ablation on other organs.

of trabecular to compact myocardium thickness between control and ablated hearts in the LV and RV (*Figure 6—figure supplement 1E*). Finally, we analyzed cell proliferation by staining pHH3 and did not observe differences between the two conditions (*Figure 6—figure supplement 1F*). Next, we increased the tamoxifen treatments to three times, ranging from P1 to P3, and collected hearts at P4

(*Figure 6A*). The heart morphology and size did not reveal obvious differences between control and ablated groups (*Figure 6B*). The CD31 staining revealed a reduction in endothelial cell density in the ablated hearts when compared to the control hearts (*Figure 6C*). These reductions in both LV and RV were further confirmed through quantifications (*Figure 6D*). Finally, we quantified the chamber thickness and cell proliferation. Similar to hearts treated with one dose of tamoxifen, we observed no apparent defects and cell proliferation changes in ablated hearts with three days of tamoxifen treatments (*Figure 6D*, *Figure 6—figure supplement 1G*). Furthermore, we stained the P4 hearts for collagen depositions. We observed that the ablated hearts had a sparser collagen signal and lost most of the stripe pattern in the septum compared to the control hearts. However, a significant amount of stripe-patterned collagen was still observed in the ventricular free wall, although the overall signal was sparser (*Figure 6E*). These collagens were likely derived from FBs before ablation.

Lastly, we conducted a long-term analysis by administering tamoxifen to the mice from P3 to P5 and harvesting them at P17 or P18 (*Figure 6F*). We observed that approximately two-thirds of the ablated mice had perished before P17 (*Figure 6G*), and the surviving mice were significantly smaller than the control mice. Quantification revealed reduced heart rates and body weights, but a similar heart-to-body ratio in the ablated mice compared to controls (*Figure 6I*). A detailed examination of the major tissues showed reduced organ sizes in the ablated mice, including smaller hearts, lungs, brains, kidneys, and livers (*Figure 6F*, *Figure 6—figure supplement 3A*). Additionally, we assessed mouse heart function using echocardiography before sacrificing them. Interestingly, we found that heart function, including ejection fraction (EF) and fractional shortening (FS), was not reduced in the ablated mice compared to controls (*Figure 6H, I*). Furthermore, we evaluated collagen levels in the ablated hearts and did not observe a clear reduction or pattern change (*Figure 6—figure supplement 2*). This finding contrasts with our observations at P4 (*Figure 6E*) and suggests potential collagen expression compensation at later stages. In contrast, we found that the lungs of the ablated mice exhibited reduced collagen levels and larger empty spaces compared to the controls (*Figure 6—figure supplement 3B*). These findings suggest that FBs play a crucial role in the growth of multiple tissues during the neonatal phase. Additionally, it will be interesting to explore the mechanisms underlying the preservation of heart function when FBs are eliminated in future studies.

## Discussion

In this study, we investigated the cellular and molecular interactions between FB and CM or Vas_EC, identifying collagen as the primary signaling molecule. We also analyzed the role of FB in embryonic heart development using a cell ablation system. Additionally, we employed scRNA-seq to identify the molecular defects in CM and Vas_EC within the ablated hearts. Finally, we examined the function of FB in neonatal heart development through both short-term and long-term ablation experiments. Altogether, our study represents the first comprehensive analysis of FB function in heart development. The insights we gained from the study will not only enhance our understanding of the role of this major cardiac cell type in heart development but also shed light on its function in adult cardiac remodeling, as well as other tissue development and regeneration.

Through RNA staining of the three lineage genes, we found that each FB directly interacts with approximately one Vas_EC and four CMs. Given that the quantification was conducted on 2D tissue sections, the number of contacts may be underestimated. Considering that Vas_EC and FB emerged around the same time, it will be important to understand in the future how their connections are established at the single-cell level and what these connections mean for the development of each lineage. Two studies in mice and zebrafish have shown that epicardial cells or epicardium-derived cells regulate Vas_EC development through the expression of *Hapln1* or *Slit2* (*Quijada et al., 2021*; *Sun et al., 2022*). It would be interesting to test whether collagen or other FB-derived ECM proteins have similar roles. Additionally, in vitro culture experiments have demonstrated that FBs interact with CMs through gap junction proteins or integrins to regulate their proliferation (*Giacomelli et al., 2020*; *Ieda et al., 2009*). Future studies should aim to clarify how these interactions are established in vivo and how they evolve during developmental progression.

Collagen, as the major ECM family, consists of 28 members (*Frantz et al., 2010*). In addition to collagen, there are many other types of ECM genes. Our scRNA-seq data revealed the temporal and spatial expression patterns of ECM genes in cardiac FBs. Future research should investigate their differential functions. For example, we found that most ECM genes in group 4 upregulate their expression

by postnatal day 7, when mouse hearts begin to lose their regenerative capacity, suggesting a potential role in heart regeneration. This warrants further investigation. Additionally, while we have analyzed the collagen pathway as a whole in this study, our scRNA-seq analysis indicated that collagen genes in different groups exhibit distinct expression patterns. It will be essential to explore their unique roles in heart development in future studies. Furthermore, we found that collagen is the primary signaling molecule in FB_CM and FB_VasEC pairs in mice; however, we only observed it in the FB_VasEC pair in humans. This discrepancy may be due to the loss of collagen receptors in human CMs, which merits further exploration.

The DTA-based cell ablation system has been used to ablate cardiac progenitors and cardiomyocytes before. It revealed that a robust regeneration response occurred after cell ablation at early stages (*Sturzu et al., 2015*). However, in our study, based on the pHH3 staining, we did not observe an increase in cell proliferation after cell ablation. This could be due to multiple possibilities. First, the *Pdgfra-CreER; Rosa-DTA*-based ablation may be too severe to initiate the response to cell loss. Considering that ablation at E13.5 and ablation with three doses of tamoxifen at E15.5 both led to smaller embryos, the ablation of a large number of cells may compromise the compensation mechanisms. However, it is difficult to explain the embryos at E15.5 with one dose of tamoxifen treatment, which did not display obvious defects after ablation and have no increase in cell proliferation. Second, unlike cardiomyocytes, FBs may have a limited compensatory response to their own ablation, although they can respond to CM loss at both neonatal and adult stages.

ScRNA-seq results revealed the upregulation of inflammation-related pathways in Vas_EC in the ablated hearts. It will be important to investigate whether these pathways are initiated by the ablation process or the loss of FBs in the future. Given that these pathways were primarily upregulated in Vas_EC but not in CM, this suggests a cell-type-specific response. Studying the function of these signals in vascular development will be essential. Moreover, pathway analysis revealed both upregulation and downregulation of signaling pathways associated with dying FBs. Further studies on these pathways are important, as they could offer valuable insights into the cell death process during normal heart development and heart injury and repair.

FB ablation at the neonatal stage led to two-thirds of the mice experiencing a lethal outcome, while the remaining mice developed a dwarf phenotype. Functional analysis of the mouse hearts showed preserved cardiac function post-ablation (*Figure 6H, I*), suggesting that the lethal phenotype may be due to defects in other tissues, such as the lungs, which exhibited disrupted structures.

## Methods
### Experimental methods
#### Mouse strains

The animal experiments have been approved by the University of Pittsburgh Institutional Animal Care and Use Committee (IACUC). CD1 male and female mice were purchased from Charles River Laboratories and bred in our laboratory to generate embryos and neonatal pups at specific stages for RNA staining experiments. The transgenic mice, including *Rosa26-mTmG* (Strain #:007676) (*Muzumdar et al., 2007*), *Pdgfra-CreERT2* (Strain #:032770) (*Chung et al., 2018*), *Postn-CreER* (Strain #:029645) (*Kanisicak et al., 2016*), and *ROSA26-eGFP-DTA* (Strain #:032087) (*Ivanova et al., 2005*), were ordered from the Jackson Laboratory.

#### Tamoxifen treatment and mouse dissection

The male and female mice of specific strains were bred together. The default dosage of 200 µg of tamoxifen per gram of body weight (200 µg/g) was given to pregnant mice through oral gavage, and neonatal mice were given 10 µg/g of tamoxifen by direct injection into their stomach to induce Cre activity (*Hortells et al., 2020*). The pregnant mice and neonatal pups were euthanized using CO2 and decapitation-based methods, respectively. The mouse hearts were isolated following the standard procedure described previously and were directly embedded in OCT without fixation for RNA staining or fixed at 4% paraformaldehyde for other experiments, such as immunofluorescence staining.

## Proximity ligation in situ hybridization (PLISH)

PLISH was performed following a published protocol (*Nagendran et al., 2018*). Specifically, the embryonic or postnatal mouse hearts were embedded in OCT (Sakura, 4583) without fixation. After sectioning at a thickness of 10 µm, the tissue sections were treated with post-fix medium (RNase-free PBS with 3.7% formaldehyde and 0.1% DEPC) followed by 0.1 mg/ml pepsin treatment (RNase-free H2O with 0.1 mg/ml pepsin and 0.1 M HCl). After dehydration, the sections were sealed with hybridization chambers (Invitrogen, S24732) and hybridized with H probes (*Supplementary file 1*) in Hybridization Buffer (1 M NaTCA, 5 mM EDTA, 50 mM Tris pH 7.4, 0.2 mg/mL Heparin). Next, after being treated with circularization reaction and rolling cycle amplification, the samples were hybridized with detection probes conjugated with Cy3 or Cy5 fluorophore. Finally, the samples were stained with DAPI (Invitrogen, D1306), mounted with fluoromount-g (SouthernBiotech, OB100-01), and imaged under confocal microscopy (Leica TSC SP8).

## RNAscope Multiplex Fluorescent V2 assay

The RNAscope Multiplex Fluorescent Reagent Kit v2 (Advanced Cell Diagnostics, 323270) was performed according to the manufacturer's manual. Briefly, the tissue sections were fixed in pre-chilled 4% PFA (Electron Microscopy Sciences, 15710) at 4°C for 1 hr. After progressive dehydration, the sections were sequentially treated with hydrogen peroxide for 10 min and protease IV for 15 min (for embryonic) or 20 min (for postnatal) at room temperature (RT). After that, the samples were hybridized to gene-specific Z probes for 2 hr at 40°C using the HybEZ II Hybridization system (ACD, 321721). Following further signal amplification, the hybridization signals were detected with TSA Vivid Fluorophores. The samples were then stained with DAPI and mounted with ProLong Gold Antifade Mountant (Invitrogen, P36930), and imaged under confocal microscopy (Leica TSC SP8). The RNAscope probes used in the study are Mm-Col1a1-C2 (319371-C2), Mm-Actn2 (569061), and Mm-Cdh5-C3 (312531-C3).

## Quantification of the interacting cells

Based on the RNA scope staining results, the direct contacts between FBs and cardiomyocytes as well as endothelial cells were quantified. Given that the FB starts to emerge at E13.5, we mainly quantified the contacts at E17.5 and P3. Approximately 100 FBs from the left ventricular wall at each stage were selected, and the number of $Actn2^+$ and $Cdh5^+$ cells surrounding each FB was counted, respectively.

## Immunofluorescence staining

The immunofluorescence staining was performed following a standard procedure. Briefly, mouse hearts were fixed in 4% PFA overnight before embedding in OCT. Afterwards, the samples were sectioned at 10 µm and briefly washed in PBS to clean the OCT. The sections were then blocked for 1 hr in blocking buffer (10% goat serum, 1% BSA, 0.1% Tween 20) and incubated with primary antibodies in the primary antibody buffer (1% BSA in PBST) at 4°C overnight. On the second day, the samples were stained with fluorophore-conjugated secondary antibodies in blocking buffer for 1 hr at RT. Finally, the samples were stained with DAPI, mounted with fluoromount-g, and imaged with a confocal microscope. The primary antibodies used in the study include anti-CD31 (BD, # 550274), and anti-pHH3-488 (abcam, #ab197502). The secondary antibodies used are Goat Anti-Rabbit IgG-488 (Thermo Fisher, # A-11008) and Goat Anti-Rabbit IgG-647 (Thermo Fisher, #A21247).

## Collagen staining

Fresh frozen sections from embryonic and postnatal mouse hearts were fixed in 4% PFA in 1 X PBS for 15 min at RT. After that, the sections were soaked in a sodium citrate-based solution (distilled water with 10 mM sodium citrate and 0.5% Tween-20, HCl was added to adjust pH to 6.0) for antigen retrieval, which was performed in a steamer for 30 min. Subsequently, the sections were incubated with 20 µM biotin-conjugated collagen hybridizing peptide (Advanced Biomatrix, 50-196-0307) in 1 X PBS at 4°C overnight. The next day, streptavidin-cy5 (Invitrogen, SA1011) was used (1:500 in 1 X PBS with 1% BSA) for 1 hr at RT. Finally, the samples were stained with DAPI, mounted with FLUORO-MOUNT-G, and imaged under a confocal microscope.

## TUNEL staining

TUNEL staining was performed following the manufacturer's protocol (Roche, 11684795910). Briefly, mouse hearts were embedded and sectioned as described in the immunofluorescence staining procedure. After fixation with 4% paraformaldehyde in PBS for 20 min and three washes with PBS, the sections were incubated in permeabilization solution (0.1% Triton X-100, 0.1% sodium citrate) for 2 min on ice, followed by two PBS rinses. Subsequently, 60 µl of TUNEL reaction mixture was added to each section, and they were incubated in a humidified atmosphere for 1 hr at 37°C in the dark. Finally, the sections were washed with PBS three times, stained with DAPI, mounted with fluoromount-g, and imaged with a confocal microscope.

## Single-cell mRNA-sequencing experiments

ScRNA-seq was performed following the MULTI-seq procedure previously reported (*McGinnis et al., 2019*; *Feng et al., 2022*). Briefly, tamoxifen-treated E16.5 and E18.5 Pdgfra-CreER; Rosa-DTA mouse embryos were harvested on the same day. Since the ablated mouse embryos at both stages were smaller than the control embryos, we selected one control and one ablated embryo from each stage for heart dissection. The genotype of the selected embryos was further validated using PCR. The hearts, with four chambers, were dissociated into single cells and labeled with MULTI-seq barcodes. The pooled libraries were then loaded onto the 10 X Genomics Chromium iX and profiled using the single-cell 3' V3.1 kit. The generated libraries were sequenced on the Illumina NovaSeq X Plus platform. The experiments were repeated twice.

## Echo imaging

Echocardiography was performed using a standard protocol (*Zhang et al., 2022*). Briefly, heart function of awake mice at P17 was assessed using the Vevo 3100 micro imaging platform (FUJIFILM Visual Sonics Inc, Canada). The measurements were conducted at the level of the papillary muscle in M mode. Heart rates, ejection fraction (EF), and fractional shortening (FS) were calculated from the 2D short-axis view.

## Statistical analysis

To quantify pHH3 signal, compact, and trabecular myocardium thickness, we used two to three sections from each heart and more than three hearts per genotype. The exact number of embryos profiled can be found in each figure legend. Statistical analyses were conducted using ImageJ and Prism 9 software, employing a two-tailed Student's t-test to compare groups. p-Values below 0.05 were considered significant.

## **Data analysis**

### ScRNA-seq data analysis

ScRNA-seq data were mapped to the mouse genome mm10 using CellRanger and further de-multiplexed with the R package deMULTIplex (*McGinnis et al., 2019*). Cell types in the newly generated scRNA-seq data were annotated using lineage genes previously published by our team (*Feng et al., 2022*). We performed quality control by removing cells with fewer than 200 genes and those with more than 40% mitochondrial content. Experimental batch integration was conducted using Harmony 1.2.0 (*Korsunsky et al., 2019*), followed by unsupervised clustering with Seurat V5 (*Hao et al., 2024*). Next, we utilized the Cell Cycle Scoring function in Seurat to annotate the cell cycle phases for each individual cell. Finally, we applied the FindMarkers function in Seurat with default settings to identify genes that were differentially expressed in Ven_CM or Vas_EC under control and ablation conditions.

### Extracellular matrix genes expression analysis

We downloaded the extracellular matrix genes from the Jackson Laboratory under the term 'extracellular matrix' and gene ontology ID 0031012 (https://www.informatics.jax.org/go/term/GO:0031012; Download date: 2023-07-07). After downloading, we cleaned the gene list by removing duplicates and filtering out genes with null expression, resulting in a total of 440 genes. Next, we assessed the enrichment of each gene in the main population of FBs at each stage and chamber using the R

package AUCell (*Aibar et al., 2017*), with heatmaps drawn with the R package ggplot2 (*Figure 2— figure supplements 2–4*).

## Ligand-receptor interaction analysis

To identify ligand-receptor interactions across stages and zones for FBs and cardiomyocytes, we analyzed relevant subsets of the CD1 data, including the main population of FB, ventricular CM, and Vas_EC using CellPhoneDB v3.34 (*Efremova et al., 2020*). We used a p-value threshold of 0.2 and ran the analysis with 10 threads, while keeping the rest of the parameters at their default settings. The significant mean value of all interactive partners (log2) and enrichment p-values (-log10) obtained from the CellPhoneDB outputs were plotted as dot plots in R.

## CellChat analysis

The Seurat object containing mouse CD1 single-cell mRNA sequencing data was converted into a CellChat object in R. Default settings in CellChat 1.6.1 (*Jin et al., 2021*) were used to identify the number and strength of interactions among cell types. FB were designated as sender cells, while Ven_CM and Vas_EC were designated as receiver cells to identify signaling between these related cell types. The same default settings were applied to analyze signaling pathways among human cardiac cell types. For pathway analysis in control and DTA-ablated cardiac cells, their respective scRNA-seq objects were utilized.

## NicheNet analysis

Nichenetr 2.1.5 (*Browaeys et al., 2020*) was used for the analysis with default settings. First, we identified differentially expressed genes between control and ablated cells. Due to the varying number of recovered cells in each cell type under different conditions, we selected different staged samples for comparison and assessed their consistency across the remaining samples. For Ven_CMs, we used the E16.5_control_1 and E16.5_DTA_1 samples, while for Vas_EC, we included all four E16.5 samples. Next, we designated the control sample as the background condition and the ablated sample as the experimental condition, using main_fb as the sending cells and Ven_CM or Vas_EC as the receiving cells to predict their regulatory potential.

# Acknowledgements

We'd like to thank all the members in the Li laboratory for their insightful discussions of this work. We are thankful to Dr. Wei Feng for his help on the MULTI-seq experiment. This research was supported in part by the University of Pittsburgh Center for Research Computing, RRID:SCR_022735, through the resources provided. Specifically, this work used the HTC cluster, which is supported by NIH award number S10OD028483. This work was supported by R00HL133472 and DP2HL163745 from the NIH, the SVRF grant from Additional Ventures, and the CMRF grant from the University of Pittsburgh.

# Additional information

## Funding

| Funder | Grant reference number | Author |
| --- | --- | --- |
| NIH Office of the Director | DP2HL163745 | Guang Li |
| National Heart, Lung, and Blood Institute | R00HL133472 | Guang Li |
| Additional Ventures | SVRF grant | Guang Li |

The funders had no role in study design, data collection and interpretation, or the decision to submit the work for publication.

## Author contributions

Yiting Deng, Yuanhang He, Conceptualization, Formal analysis, Validation, Investigation, Methodology, Writing – original draft, Writing – review and editing; Juan Xu, Manling Zhang, Investigation, Methodology, Writing – review and editing; Haoting He, Software, Investigation, Methodology, Writing – original draft, Writing – review and editing; Guang Li, Conceptualization, Data curation, Formal analysis, Supervision, Funding acquisition, Investigation, Methodology, Writing – original draft, Project administration, Writing – review and editing

## Author ORCIDs

Juan Xu [iD] https://orcid.org/0009-0004-3184-8730
Guang Li [iD] https://orcid.org/0000-0002-8546-2364

Reviewer #1 (Public review): https://doi.org/10.7554/eLife.102305.4.sa1
Reviewer #2 (Public review): https://doi.org/10.7554/eLife.102305.4.sa2
Reviewer #3 (Public review): https://doi.org/10.7554/eLife.102305.4.sa3
Author response https://doi.org/10.7554/eLife.102305.4.sa4

---

# Additional files

## Supplementary files

Supplementary file 1. Primer sequences for PLISH probes.

Supplementary file 2. The interaction strength values among the different cell types in *Figure 2A*.

Supplementary file 3. The enrichment score of extracellular matrix genes in CD1 main population fibroblasts.

Supplementary file 4. The enrichment score of extracellular matrix genes in C57BL/6 main population fibroblasts.

Supplementary file 5. The list of ligand-receptor interactions between the main population of fibroblast and ventricular cardiomycocytes in CD1 dataset.

Supplementary file 6. The list of ligand-receptor interactions between the main population of fibroblast and Vas_EC in CD1 dataset.

Supplementary file 7. The list of genes that differentially expressed in control and ablated Ven_CMs at e16.5 stage.

Supplementary file 8. The list of genes that differentially expressed in control and ablated Ven_CMs at e18.5 stage.

Supplementary file 9. The list of genes that differentially expressed in control and ablated Vas_ECs.

MDAR checklist

## Data availability

The newly generated scRNA-seq datasets have been deposited in the Gene Expression Omnibus (GEO) under the accession number GSE272048. The scRNA-seq datasets from CD1 and C57BL/6 mouse hearts were generated in a previous study and can be downloaded from the GEO database using the accession number GSE19334636 (*Feng et al., 2022*). The human scRNA-seq data were obtained from GEO or the European Genome-Phenome Archive (EGA) under the accessions GSE106118 and EGAS0000100399627-29 (*Asp et al., 2019*; *Suryawanshi et al., 2020*; *Cui et al., 2019*) for CellChat analysis.

The following dataset was generated:

| Author(s) | Year | Dataset title | Dataset URL | Database and Identifier |
|---|---|---|---|---|
| Deng Y, He H, Li G | 2024 | single cell transcriptomic analysis of embyronic hearts with fibroblast ablation | https://www.ncbi.nlm.nih.gov/geo/query/acc.cgi?acc=GSE272048 | NCBI Gene Expression Omnibus, GSE272048 |

The following previously published datasets were used:

| Author(s) | Year | Dataset title | Dataset URL | Database and Identifier |
|---|---|---|---|---|
| Asp M, Giacomello S, Larsson L, Wu C, Fürth D, Qian X, Wärdell E, Custodio J, Reimegård J, Salmén F, Österholm C, Ståhl P, Sundström E, Åkesson E, Bergmann O, Bienko M, Månsson-Broberg A, Nilsson M, Sylvén C, Lundeberg J | 2019 | A Spatiotemporal Organ-Wide Gene Expression and Cell Atlas of the Developing Human Heart | https://ega-archive.org/datasets/EGAD00001005468 | European Genome-Phenome Archive (EGA), EGAS00001003996 |
| Feng W, Bais A, He H, Rios C, Jiang S, Xu J, Chang C, Kostka D, Li G | 2022 | Single-cell transcriptomic analysis identifies murine heart molecular features at embryonic and neonatal stages | https://www.ncbi.nlm.nih.gov/geo/query/acc.cgi?acc=GSE193346 | NCBI Gene Expression Omnibus, GSE193346 |
| Cui Y, Zheng Y, Liu X, Yan L, Fan X, Yong J, Hu Y, Dong J, Li Q, Wu X, Gao S, Li J, Wen L, Qiao J, Tang F | 2019 | Single-Cell Transcriptome Analysis Maps the Developmental Track of the Human Heart | https://www.ncbi.nlm.nih.gov/geo/query/acc.cgi?acc=GSE106118 | NCBI Gene Expression Omnibus, GSE106118 |

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
