## [Editor Report · eLife Assessment]

This study provides a comprehensive analysis of gene expression and bioinformatics data, offering **important** insights into the roles of fibroblasts in cardiac development. The large and well-analyzed single-cell RNA sequencing (scRNA-seq) dataset is **compelling** and a significant contribution to the field, and will be of broad interest to the scientific community.

---

## [Referee Report · Reviewer #1 (Public review)]

Summary:

The study by Deng et al reports single cell expression analysis of developing mouse hearts and examines the requirements for cardiac fibroblasts in heart maturation. The work includes extensive gene expression profiling and bioinformatic analysis. The prenatal fibroblast ablation studies show new information on the requirement of these cells on heart maturation before birth.

The strengths of the manuscript are the new single cell datasets and comprehensive approach to ablating cardiac fibroblasts in pre and postnatal development in mice. Extensive data are presented on mouse embryo fibroblast diversity and morphology in response to fibroblast ablation. Histological data support localization of major cardiac cell types and effects of fibroblast ablation on cardiac gene expression at different times of development.

A weakness of the study is that the major conclusions regarding collagen signaling and heart maturation are based on gene expression patterns and are not functionally validated.

Comments on Revised Version (from BRE):

Most of my comments have been adequately addressed. Additional comments on new data in the revised manuscript are below.

(1) In the new figure S11, it is not really possible to draw major conclusions on mitral valve morphology and maturation since the planes of sections to not seem comparable. Observations regarding attachment to the papillary muscle might be dependent on the particular section being evaluated. However, it is useful to see that the valves are not severely affected in the ablated animals.

(2) In the last supplemental figure S19, it is not possible to determine if results are or are not statistically significant for n=2 as shown for FS and EF for the ablated animals and controls. The text says that there is a trend of improved heart function, but evaluation of additional animals is needed to support this conclusion.

---

## [Referee Report · Reviewer #2 (Public review)]

This study aims to elucidate the role of fibroblasts in regulating myocardium and vascular development through signaling to cardiomyocytes and endothelial cells. This focus is significant, given that fibroblasts, cardiomyocytes, and vascular endothelial cells are the three primary cell types in the heart. The authors employed a Pdgfra-CreER-controlled diphtheria toxin A (DTA) system to ablate fibroblasts at various embryonic and postnatal stages, characterizing the resulting cardiac defects, particularly in myocardium and vasculature development. Single-cell RNA sequencing (scRNA-seq) analysis of the ablated hearts identified collagen as a crucial signaling molecule from fibroblasts that influences the development of cardiomyocytes and vascular endothelial cells.

This is an interesting manuscript; however, there are several major issues, including an over-reliance on the scRNA-seq data, which shows inconsistencies between replicates.

Some of the major issues are described below.

(1) The CD31 immunostaining data (Figure 3B-G) indicate a reduction in endothelial cell numbers following fibroblast deletion using PdgfraCreER+/-; RosaDTA+/- mice. However, the scRNA-seq data show no percentage change in the endothelial cell population (Figure 4D). Furthermore, while the percentage of Vas_ECs decreased in ablated samples at E16.5, the results at E18.5 were inconsistent, showing an increase in one replicate and a decrease in another, raising concerns about the reliability of the RNA-seq findings.

(2) Similarly, while the percentage of Ven_CMs increased at E18.5, it exhibited differing trends at E16.5 (Fig. 4E), further highlighting the inconsistency of the scRNA-seq analysis with the other data.

(3) Furthermore, the authors noted that the ablated samples had slightly higher percentages of cardiomyocytes in the G1 phase compared to controls (Fig. 4H, S11D), which aligns with the enrichment of pathways related to heart development, sarcomere organization, heart tube morphogenesis, and cell proliferation. However, it is unclear how this correlates with heart development, given that the hearts of ablated mice are significantly smaller than those of controls (Figure 3E). Additionally, the heart sections from ablated samples used for CD31/DAPI staining in Figure 3F appear much larger than those of the controls, raising further inconsistencies in the manuscript.

(4) The manuscript relies heavily on the scRNA-seq dataset, which shows inconsistencies between the two replicates. Furthermore, the morphological and histological analyses do not align with the scRNA-seq findings.

(5) There is a lack of mechanistic insight into how collagen, as a key signaling molecule from fibroblasts, affects the development of cardiomyocytes and vascular endothelial cells.

(6) In Figure 1B, Col1a1 expression is observed in the epicardial cells (Figure 1A, E11.5), but this is not represented in the accompanying cartoon.

(7) Do the PdgfraCreER+/-; RosaDTA+/- mice survive after birth when induced at E15.5, and do they exhibit any cardiac defects?

Comments on Revised Version (from BRE):

The manuscript has greatly improved following the revision, and I have no additional comments to offer.

---

## [Referee Report · Reviewer #3 (Public review)]

Summary:

The authors investigated fibroblasts' communication with key cell types in developing and neonatal hearts, with focus on critical roles of fibroblast-cardiomyocyte and fibroblast-endothelial cells network in cardiac morphogenesis. They tried to map the spatial distribution of these cell types and reported the major pathways and signaling molecules driving the communication. They also used Cre-DTA system to ablate Pdgfra labeled cells and observed myocardial and endothelial cell defects at development. They screened the pathways and genes using sequencing data of ablated heart. Lastly they reported a compensatory collagen expression in long term ablated neonate heart. Overall, this study provides us with important insight on fibroblasts' roles in cardiac development and will be a powerful resource for collagens and ECM focused research.

Strengths:

The authors utilized good analyzing tools to investigate on multiple database of single cell sequencing and Multi-seq. They identified significant pathways, cellular and molecular interactions of fibroblasts. Additionally, they compared some of their analytic findings with human database, and identified several groups of ECM genes with varying roles in mice.

Weaknesses:

This study is majorly based on sequencing data analysis. At the bench, they used very strident technique to study fibroblast functions by ablating one of the major cell population of heart. Also, experimental validation of their analyzed downstream pathways will be required eventually.

Comments on Revised Version (from BRE):

The authors did a good job addressing the questions asked at first review. However, I have some minor concerns.

(1) The paper notes that collagen signaling is observed in FB-VasEC in humans, but not in FB-VenCM, unlike mice. Did the authors analyze predictive ligand receptor interaction as they did with control and ablated mice heart? This could add valuable new insights that how FB regulate ventricular CM in human heart.

(2) The authors provided data on Defect in CD31 expression in several models. Did they observe any other phenotypes associated with defective endothelial or vascular system? Such as, blood accumulation in pericardium, larger/smaller capillaries? Did they also examine percentage of Cdh5+ cells?

(3) Please mention the sample age of Figure 2A-C.

(4) Please follow the same style to describe X axis in graphs in Figure 3D (and all similar graphs in the manuscript) as followed in 3G.

(5) It is important to provide echocardiographic M mode images with a comparable number of cardiac cycles in control and ablated (Fig. 6H).

(6) In the long-term neonatal ablation experiments, collagen expressions return to normal. The manuscript attributes this to possible "compensatory expression," Do they have any thoughts how this is regulated? Are other cell types stepping in, or are surviving FBs proliferating?

(7) While collagen is shown to be a dominant signaling molecule, its centrality is inferred primarily from scRNA-seq and ligand-receptor predictions. Did authors try any functional rescue experiment (e.g., exogenous collagen supplementation or receptor blockade) to directly validate this pathway's role in vivo?

---

## [Author Response]

The following is the authors’ response to the previous reviews.

**Public Reviews:**

**Reviewer #1 (Public review):**
Summary:The study by Deng et al reports single cell expression analysis of developing mouse hearts and examines the requirements for cardiac fibroblasts in heart maturation. The work includes extensive gene expression profiling and bioinformatic analysis. The prenatal fibroblast ablation studies show new information on the requirement of these cells on heart maturation before birth.The strengths of the manuscript are the new single cell datasets and comprehensive approach to ablating cardiac fibroblasts in pre and postnatal development in mice. Extensive data are presented on mouse embryo fibroblast diversity and morphology in response to fibroblast ablation. Histological data support localization of major cardiac cell types and effects of fibroblast ablation on cardiac gene expression at different times of development.A weakness of the study is that the major conclusions regarding collagen signaling and heart maturation are based on gene expression patterns and are not functionally validated.
**Reviewer #2 (Public review):**
This study aims to elucidate the role of fibroblasts in regulating myocardium and vascular development through signaling to cardiomyocytes and endothelial cells. This focus is significant, given that fibroblasts, cardiomyocytes, and vascular endothelial cells are the three primary cell types in the heart. The authors employed a Pdgfra-CreER-controlled diphtheria toxin A (DTA) system to ablate fibroblasts at various embryonic and postnatal stages, characterizing the resulting cardiac defects, particularly in myocardium and vasculature development. Single-cell RNA sequencing (scRNA-seq) analysis of the ablated hearts identified collagen as a crucial signaling molecule from fibroblasts that influences the development of cardiomyocytes and vascular endothelial cells.This is an interesting manuscript; however, there are several major issues, including an over-reliance on the scRNA-seq data, which shows inconsistencies between replicates.

We thank the reviewer for carefully reading our revised manuscript. All of the questions listed below were raised in the previous round and have been addressed in the current revision. As noted in the “Recommendations for the Authors” section, the reviewer has no additional comments at this time.

Some of the major issues are described below.(1) The CD31 immunostaining data (Figure 3B-G) indicate a reduction in endothelial cell numbers following fibroblast deletion using PdgfraCreER+/-; RosaDTA+/- mice. However, the scRNA-seq data show no percentage change in the endothelial cell population (Figure 4D). Furthermore, while the percentage of Vas_ECs decreased in ablated samples at E16.5, the results at E18.5 were inconsistent, showing an increase in one replicate and a decrease in another, raising concerns about the reliability of the RNA-seq findings.(2) Similarly, while the percentage of Ven_CMs increased at E18.5, it exhibited differing trends at E16.5 (Fig. 4E), further highlighting the inconsistency of the scRNA-seq analysis with the other data.(3) Furthermore, the authors noted that the ablated samples had slightly higher percentages of cardiomyocytes in the G1 phase compared to controls (Fig. 4H, S11D), which aligns with the enrichment of pathways related to heart development, sarcomere organization, heart tube morphogenesis, and cell proliferation. However, it is unclear how this correlates with heart development, given that the hearts of ablated mice are significantly smaller than those of controls (Figure 3E). Additionally, the heart sections from ablated samples used for CD31/DAPI staining in Figure 3F appear much larger than those of the controls, raising further inconsistencies in the manuscript.(4) The manuscript relies heavily on the scRNA-seq dataset, which shows inconsistencies between the two replicates. Furthermore, the morphological and histological analyses do not align with the scRNA-seq findings.(5) There is a lack of mechanistic insight into how collagen, as a key signaling molecule from fibroblasts, affects the development of cardiomyocytes and vascular endothelial cells.(6) In Figure 1B, Col1a1 expression is observed in the epicardial cells (Figure 1A, E11.5), but this is not represented in the accompanying cartoon.(7) Do the PdgfraCreER+/-; RosaDTA+/- mice survive after birth when induced at E15.5, and do they exhibit any cardiac defects?
**Reviewer #3 (Public review):**
Summary:The authors investigated fibroblasts' communication with key cell types in developing and neonatal hearts, with focus on critical roles of fibroblast-cardiomyocyte and fibroblast-endothelial cells network in cardiac morphogenesis. They tried to map the spatial distribution of these cell types and reported the major pathways and signaling molecules driving the communication. They also used Cre-DTA system to ablate Pdgfra labeled cells and observed myocardial and endothelial cell defects at development. They screened the pathways and genes using sequencing data of ablated heart. Lastly they reported a compensatory collagen expression in long term ablated neonate heart. Overall, this study provides us with important insight on fibroblasts' roles in cardiac development and will be a powerful resource for collagens and ECM focused research.Strengths:The authors utilized good analyzing tools to investigate on multiple database of single cell sequencing and Multi-seq. They identified significant pathways, cellular and molecular interactions of fibroblasts. Additionally, they compared some of their analytic findings with human database, and identified several groups of ECM genes with varying roles in mice.Weaknesses:This study is majorly based on sequencing data analysis. At the bench, they used very strident technique to study fibroblast functions by ablating one of the major cell population of heart. Also, experimental validation of their analyzed downstream pathways will be required eventually.
**Recommendations for the authors:**

**Reviewer #1 (Recommendations for the authors):**
Most of my comments have been adequately addressed. Additional comments on new data in the revised manuscript are below.(1) In the new figure S11, it is not really possible to draw major conclusions on mitral valve morphology and maturation since the planes of sections to not seem comparable. Observations regarding attachment to the papillary muscle might be dependent on the particular section being evaluated. However, it is useful to see that the valves are not severely affected in the ablated animals.

We appreciate the reviewer’s comment and agree with the reviewer’s observation. Accordingly, we have updated the manuscript by removing the original conclusion-related statement and instead highlighting that the valves were not severely affected in the ablated animals (page 6).

(2) In the last supplemental figure S19, it is not possible to determine if results are or are not statistically significant for n=2 as shown for FS and EF for the ablated animals and controls. The text says that there is a trend of improved heart function, but evaluation of additional animals is needed to support this conclusion.

We thank the reviewer for the comment and agree that a sample size of n = 2 is too small to draw meaningful conclusions. As previously suggested by the reviewer, we have removed this result from the manuscript (page 10).

**Reviewer #2 (Recommendations for the authors):**
The manuscript has greatly improved following the revision, and I have no additional comments to offer.

Thanks!

**Reviewer #3 (Recommendations for the authors):**
Authors did a good job addressing questions asked at first review. However, I have some minor concerns.(1) The paper notes that collagen signaling is observed in FB-VasEC in humans, but not in FB-VenCM, unlike mice. Did authors analyze predictive ligand receptor interaction as they did with control and ablated mice heart? This could add valuable new insights that how FB regulate ventricular CM in human heart.

Thank you. We have analyzed the predicted ligand-receptor interactions between Fb and Ven_CM, as well as between Fb and Vas_EC, using human scRNA-seq data. The results are provided as a supplemental figure (Fig. S8C).

(2) The authors provided data on Defect in CD31 expression in several models. Did they observed any other phenotypes associated with defective endothelial or vascular system? Such as, blood accumulation in pericardium, larger/smaller capillaries? Did they also examined percentage of Cdh5+ cells?

We thank the reviewer for the questions. We did not observe clear evidence of blood accumulation in the pericardium of the ablated hearts, as shown in figure 3B, 3E, 6B, and 6F. Additionally, we did not perform Cdh5 staining in either the control or ablated hearts.

(3) Please mention the sample age of Figure 2A-C.

These are single-cell mRNA sequencing data from CD1 mice across 18 developmental stages, ranging from E9.5 to P9. We have added this information to the manuscript (page 4).

(4) Please follow the same style to describe X axis in graphs in Figure 3D (and all similar graphs in manuscript) as followed in 3G.

Thank you. We assume the reviewer was referring to the descriptions in the relevant figure legends. We have updated the legend for Figure 3D to ensure consistency with the description provided for Figure 3G (page 15).

(5) It is important to provide echocardiographic M mode images with a comparable number of cardiac cycles in control and ablated (Fig. 6H).

We thank the reviewer for the comment. As explained in our previous response, the echocardiographic data for both control and mutant mice were collected in conscious animals. The differences in their cardiac cycles reflect variations in heart rate, which represent a disease phenotype and cannot be altered. Therefore, we are unable to provide M-mode images with a similar number of cardiac cycles for control and ablated mice.

(6) In the long-term neonatal ablation experiments, collagen expressions return to normal. The manuscript attributes this to possible "compensatory expression," Do they have any thoughts how this is regulated? Are other cell types stepping in, or are surviving FBs proliferating?

We thank the reviewer for the question. As suggested, the compensatory collagen expression could be driven by surviving fibroblasts or other cell types. Since we currently lack evidence to exclude either possibility, we believe both could be contributing factors.

(7) While collagen is shown to be a dominant signaling molecule, its centrality is inferred primarily from scRNAseq and ligand-receptor predictions. Did authors try any functional rescue experiment (e.g., exogenous collagen supplementation or receptor blockade) to directly validate this pathway's role in vivo?

We thank the reviewer for the comment. As noted in our previous revision in response to similar questions from the other two reviewers, we agree that these rescue experiments are of interest but are beyond the scope of the current study. We plan to pursue these investigations in future work and share our findings when available.